# On the Adversarial Robustness of Camera-based 3D Object Detection

**Shaoyuan Xie** *shaoyux@uci.edu*
*Department of Computer Science*
*University of California, Irvine*

**Zichao Li** *zli489@ucsc.edu*
*Department of Computer Science and Engineering*
*University of California, Santa Cruz*

**Zeyu Wang** *zwang615@ucsc.edu*
*Department of Computer Science and Engineering*
*University of California, Santa Cruz*

**Cihang Xie** *cixie@ucsc.edu*
*Department of Computer Science and Engineering*
*University of California, Santa Cruz*

**Reviewed on OpenReview:** *https: // openreview. net/ forum? id= 6SofFlwhEv*

## Abstract

In recent years, camera-based 3D object detection has gained widespread attention for its ability to achieve high performance with low computational cost. However, the robustness of these methods to adversarial attacks has not been thoroughly examined, especially when considering their deployment in safety-critical domains like autonomous driving. In this study, we conduct the first comprehensive investigation of the robustness of leading camera-based 3D object detection approaches under various adversarial conditions. We systematically analyze the resilience of these models under two attack settings: white-box and black-box; focusing on two primary objectives: classification and localization. Additionally, we delve into two types of adversarial attack techniques: pixel-based and patch-based. Our experiments yield four interesting findings: (a) bird's-eye-view-based representations exhibit stronger robustness against localization attacks; (b) depth-estimation-free approaches have the potential to show stronger robustness; (c) accurate depth estimation effectively improves robustness for depth-estimation-based methods; (d) incorporating multi-frame benign inputs can effectively mitigate adversarial attacks. We hope our findings can steer the development of future camera-based object detection models with enhanced adversarial robustness. The code is available at: `https://github.com/Daniel-xsy/BEV-Attack`.

## 1 Introduction

Deep neural network-based 3D object detectors (Li et al., 2022b; Wang et al., 2022b; Huang et al., 2021; Liu et al., 2022a; Wang et al., 2022a; 2021; Lang et al., 2019; Vora et al., 2020; Zhou & Tuzel, 2018; Yan et al., 2018) have demonstrated promising performance across multiple challenging real-world benchmarks, including the KITTI (Geiger et al., 2012), nuScenes (Caesar et al., 2020) and Waymo Open Dataset (Sun et al., 2020). These popular approaches utilize either point clouds (*i.e.*, LiDAR-based methods) (Lang et al., 2019; Vora et al., 2020; Zhou & Tuzel, 2018; Yan et al., 2018) or images (*i.e.*, camera-based methods) (Wang et al., 2021; 2022a;b; Li et al., 2022b;a; Huang et al., 2021; Liu et al., 2022a) as their inputs for object detection. Compared to LiDAR-based methods, camera-based approaches have garnered significant attention

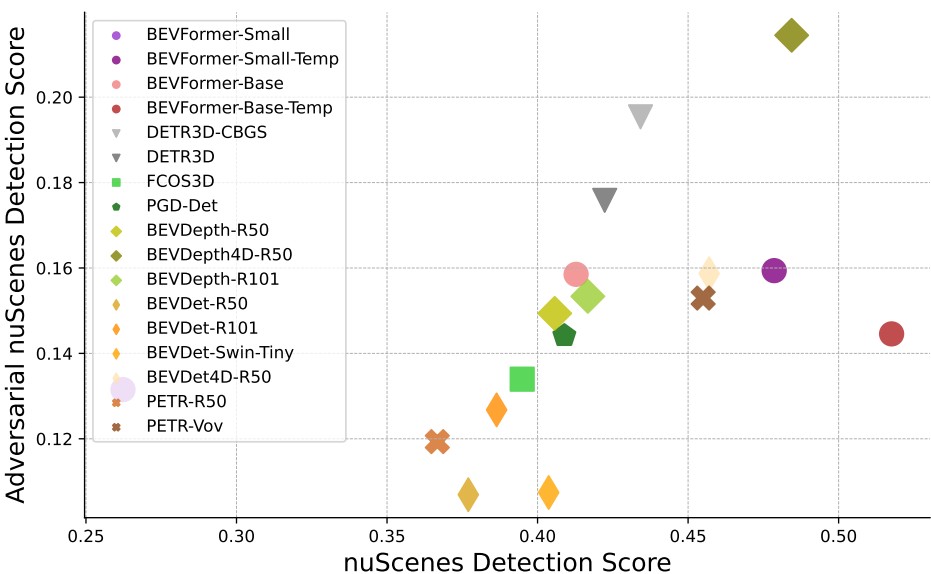

Figure 1: Adversarial nuScenes Detection Score (NDS) *v.s.* clean nuScenes Detection Score. Models that exhibit better performance on standard datasets do not necessarily exhibit better adversarial robustness.

due to their low deployment cost, high computational efficiency, and dense semantic information. Additionally, camera-based detection exhibits inherent advantages in detecting long-range objects and identifying vision-based traffic signs.

Monocular 3D object detection expands the capabilities of 2D object detection to 3D scenarios using carefully designed custom adaptations (Wang et al., 2021; 2022a). However, accurately estimating depth from a single image is challenging, often hindering the efficacy of monocular 3D object detection (Wang et al., 2022a). In contrast, using the bird's eye view (BEV) representation for 3D detection offers several advantages. First, it allows for joint learning from multi-view images. Second, the BEV perspective provides a physics-interpretable method for fusing information from different sensors and time stamps (Ma et al., 2022). Third, the output space of a BEV-based approach can be easily applied to downstream tasks such as prediction and planning. Consequently, BEV-based models have demonstrated significant improvements (Li et al., 2022a;b; Huang et al., 2021; Huang & Huang, 2022).

Despite the advancements achieved in 3D object detection algorithms, recent literature (Rossolini et al., 2022; Cao et al., 2021; Tu et al., 2020) have begun to highlight their susceptibility to adversarial attacks. Such vulnerabilities pose significant safety risks, particularly when these algorithms are deployed in safety-critical applications. Nevertheless, existing studies primarily concentrate on generating adversarial examples in limited scenarios, thereby failing to provide a comprehensive evaluation across a broader spectrum of adversarial settings and models. Motivated by this gap in the literature, we aim to conduct a thorough and systematic analysis of the robustness of various state-of-the-art 3D object detection methods against adversarial attacks, while also investigating avenues to bolster their resilience.

Our investigation includes a spectrum of attack settings: pixel-based (introducing subtle perturbations to inputs) and patch-based (overlaying discernible adversarial designs onto inputs) adversarial examples, in white-box and black-box (whether information about the model is available to the attacker) setups. Our focus is on two main attack goals: misleading classification predictions and misleading localization predictions. Regarding pixel attacks, we apply the widely-used projected gradient descent (PGD) algorithm (Madry et al., 2017). To differentiate this attack algorithm from the 3D detection method known as Probabilistic and Geometric Depth (Wang et al., 2022a), we refer to the former as *PGD-Adv* and the latter as *PGD-Det* in the rest of the paper. To further enhance the comprehensiveness of our work, we additionally use FGSM (Goodfellow et al., 2014), C&W Attack (Carlini & Wagner, 2017) and AutoPGD Attack (Croce & Hein, 2020). Regarding patch attacks, we incorporate a gradient-descent-optimized patch (Brown et al., 2017) centrally onto the target objects, adjusting its size accordingly with the object size. Additionally, we

probe the efficacy of universal patches, known for their strong transferability across varied scenes, scales, and model architectures. Overall, our experiments interestingly show that models that perform better on standard datasets do not necessarily yield stronger adversarial robustness, as shown in Fig. 1. We distill our key findings as follows:

- BEV-based models do not exhibit stronger robustness under classification attacks. However, they tend to be more robust toward localization attacks.

- Precise depth estimation is crucial for models that rely on depth information to transform the perspective view to the bird's eye view (PV2BEV). The incorporation of explicit depth supervision during training, as well as prior knowledge of depth constraints, can lead to improved performance and stronger robustness.

- Depth-estimation-free methods have achieved state-of-the-art performance with clean inputs (Wang et al., 2022b; Li et al., 2022b; Liu et al., 2022a), we further find they have the potential to yield stronger robustness compared to depth-estimation-based ones.

- Adversarial effects can be mitigated through clean temporal information. Models utilizing multi-frame benign inputs are less likely to fail under single-frame attacks. However, it is important to note that errors can accumulate under continuous adversarial input over multiple frames.

## 2 Related Work

**Camera-based 3D object detection.** Existing camera-based 3D object detection methods can be broadly categorized into two groups: monocular-based approaches (Wang et al., 2021; 2022a) and multi-view image input bird's eye view (BEV) representation-based approaches (Li et al., 2022b; Huang et al., 2021; Huang & Huang, 2022; Li et al., 2022a; Wang et al., 2022b; Liu et al., 2022a). Monocular-based approaches, such as FCOS3D and PGD-Det (Wang et al., 2021; 2022a), extend FCOS (Tian et al., 2019) to the 3D domain through specific adaptations. BEV-based detectors perform PV2BEV and build BEV representations to conduct perception tasks. Inspired by LSS (Philion & Fidler, 2020), BEVDet (Huang et al., 2021) uses an additional depth estimation branch for the PV2BEV transformation. BEVDet4D (Huang & Huang, 2022) further improves performance by leveraging temporal information. BEVDepth (Li et al., 2022a) improves depth estimation accuracy through explicit depth supervision from point clouds.

Given that inaccurate depth estimation is the main bottleneck of the above approaches, recent works explore pipelines without an explicit depth estimation branch. DETR3D (Wang et al., 2022b) represents 3D objects as object queries and performs cross-attention using a Transformer decoder (Vaswani et al., 2017). PETR (Liu et al., 2022a;b) further improves performance by proposing 3D position-aware representations. BEVFormer (Li et al., 2022b) introduces temporal cross-attention to extract BEV representations from multi-timestamp images. While these models show consistent improvement on the standard dataset, their behaviors under adversarial attacks have not been thoroughly examined, which could raise profound concerns, especially considering their potential deployment in safety-critical applications, *e.g.*, autonomous driving.

**Adversarial attacks on classification.** Modern neural networks are susceptible to adversarial attacks (Szegedy et al., 2013; Goodfellow et al., 2014; Moosavi-Dezfooli et al., 2017), where the addition of a carefully crafted perturbation to the input can cause the network to make an incorrect prediction. (Goodfellow et al., 2014) proposes a simple and efficient method for generating adversarial examples using one-step gradient descent. (Madry et al., 2017) proposes more powerful attacks (*i.e.*, PGD-Adv) by taking multiple steps along the gradients and projecting the perturbation back onto a $L_p$ norm ball. (Moosavi-Dezfooli et al., 2017) demonstrates the existence of universal adversarial perturbations. (Brown et al., 2017) generates physical adversarial patches. (Wu et al., 2021) addresses adversarial robustness in the context of long-tailed distribution recognition tasks. In addition to developing more powerful attacks, some works focus on understanding the robustness of different neural architecture designs to attacks. (Shao et al., 2021; Bai et al., 2021) conduct extensive comparisons between CNNs and Transformers and gain insights into their adversarial robustness. Different from these works which focus on classification problems, our research pivots to the detection tasks.

**Adversarial attacks on object detection.** Adversarial attacks for object detection can target both localization and classification. In the context of 2D object detection, (Xie et al., 2017) generates adversarial examples with strong transferability by considering all targets densely. (Liu et al., 2018) propose black-box patch attacks that can compromise the performance of popular frameworks such as Faster R-CNN (Ren et al., 2015). Given the importance of safety in autonomous driving, it is vital to study the adversarial robustness of 3D object detection. (Tu et al., 2020) crafts adversarial mesh placed on top of a vehicle to bypass a LiDAR detector. (Rossolini et al., 2022) studies digital, simulated, and physical patches to mislead real-time semantic segmentation models. (Cao et al., 2021) reveals the possibility of crashing Multi-Sensor Fusion (MSF) based models by attacking all fusion sources simultaneously. Despite the above works toward designing more powerful attacks, there still lacks a comprehensive understanding of the adversarial robustness of camera-based 3D object detection. Though concurrent work (Zhu et al., 2023) also explores the adversarial robustness of 3D detectors, they study much fewer models. Our research represents a pioneering effort to systematically bridge this knowledge gap.

## 3 Camera-based 3D Object Detection

In this section, we provide an overview of the current leading approaches in camera-based 3D object detection, which can be broadly classified into three categories: monocular-based detectors, BEV detectors with depth estimation, and BEV detectors without depth estimation.

### 3.1 Monocular Approach

This line of research aims to directly predict 3D targets from an image input. We select FCOS3D (Wang et al., 2021) and PGD-Det (Wang et al., 2022a) as representative works to study their adversarial robustness. FCOS3D extends FCOS (Tian et al., 2019) to the 3D domain by transforming 3D targets to the image domain. PGD-Det further improves the performance of FCOS3D by incorporating uncertainty modeling and constructing a depth propagation graph that leverages the interdependence between instances.

### 3.2 BEV Detector with Depth Estimation

This line of work first predicts a per-pixel depth map, mapping image features to corresponding 3D locations, and subsequently predicts 3D targets in the BEV representations. Building on the success of the BEV paradigm in semantic segmentation, BEVDet (Huang et al., 2021) develops the first high-performance BEV detector based on the Lift-Splat-Shoot (LSS) view transformer (Philion & Fidler, 2020). Subsequently, BEVDet4D (Huang & Huang, 2022) introduces multi-frame fusion to improve the effectiveness of temporal cue learning. BEVDepth (Li et al., 2022a) proposes to use point cloud projection to the image plane for direct supervision to depth estimation. Note that this approach can also incorporate temporal fusion, which we refer to as BEVDepth4D. We hereby aim to evaluate the robustness of these BEV models, ranging from the most basic detector (*i.e.*, BEVDet) to spatial (Depth) and temporal (4D) extensions, against attacks.

### 3.3 BEV Detector without Depth Estimation

In this set of works, trainable sparse object queries are utilized to aggregate image features without the need for depth estimation. Representative exemplars include DETR3D (Wang et al., 2022b), which connects 2D feature extraction and 3D bounding box prediction through backward geometric projection; PETR (Liu et al., 2022a;b), which enhances 2D features with 3D position-aware representations; and BEVFormer (Li et al., 2022b), which refines BEV queries using spatial and temporal cross-attention mechanisms. These approaches claim to not suffer from inaccurate depth estimation intermediate and thus achieve superior performance. Our research takes a step further, probing the robustness of these approaches when subjected to adversarial attacks.

## 4 Generating Adversarial Examples

In this section, we present our adversarial example generation algorithms. It is essential to note that this paper primarily focuses on understanding model robustness to attacks, rather than on the development of new attack algorithms. As such, our approach adapts established 2D adversarial attacks (Xie et al., 2017; Moosavi-Dezfooli et al., 2017; Madry et al., 2017; Goodfellow et al., 2014; Carlini & Wagner, 2017; Croce & Hein, 2020) for the 3D context, incorporating essential modifications to ensure compatibility. Specifically, we consider three attack settings: pixel-based white-box attacks (Madry et al., 2017; Xie et al., 2017), patch-based white-box attacks (Liu et al., 2018), and universal patch black-box attacks (Moosavi-Dezfooli et al., 2017). In the context of pixel-based and patch-based white-box attacks, we utilize two adversarial targets, namely untargeted classification attacks, and localization attacks. For patch-based black-box attacks, we focus solely on the targeted classification. The summary of these attacks is presented in Tab. 1.

Table 1: A summary of the five different attack settings implemented to examine the model robustness.

| White-box / Black-box | Pixel / Patch | Objective |
|---|---|---|
| White-box | Pixel | Untargeted Classification |
| White-box | Pixel | Localization |
| White-box | Patch | Untargeted Classification |
| White-box | Patch | Localization |
| Black-box | Patch | Targeted Classification |

### 4.1 Pixel-based Attack

Inspired by the approach in (Xie et al., 2017), we optimize the generation of adversarial examples over a set of targets. Let $\mathbf{I} \in \mathbb{R}^{C \times H \times W}$ be an input image, comprising $N$ targets given by $T = \{t_1, t_2, t_3, ..., t_N\}$. By feeding the image $\mathbf{I}$ into 3D object detectors, we can have $n$ perception results, capturing class, 3D bounding boxes, and other attributes, represented as $f(\mathbf{I}) = \{y_1, y_2, y_3, ..., y_n\}$. Here, each $y_i$ symbolizes a discrete detection attribute such as localization, class, velocity, *etc.* We then compare these predictions with the ground truth bounding boxes $T$, establishing a match when the 2D center distances on the ground plane are under a predefined threshold, as employed in (Caesar et al., 2020). The goal of adversarial examples is to intentionally produce erroneous predictions. For instance, in classification attacks, the objective is to manipulate the model into predicting an incorrect class, denoted as $f_{cls}(\mathbf{I} + \mathbf{r}, t_i) \neq l_i$, where $f_{cls}(\mathbf{I} + \mathbf{r}, t_i)$ signifies the classification results on the $i$-th target, $l_i$ represents its ground-truth classification label, and $\mathbf{r}$ denotes the adversarial perturbation. To accomplish this, we employ untargeted attacks, aiming to maximize the cross-entropy loss:

$$\mathcal{L}_{untargeted} = -\frac{1}{N} \sum_{i=1}^{N} \sum_{j=1}^{C} f_{cls}^j(\mathbf{I} + \mathbf{r}, t_i) \log p_{ij}, \tag{1}$$

where $C$ denotes the number of classes, and $f_{cls}^j$ denotes the confidence score on $j$-th class. The adversarial perturbation $\mathbf{r}$ is optimized iteratively using PGD-Adv (Madry et al., 2017) as:

$$\mathbf{r}_{i+1} = \text{Proj}_\epsilon(\mathbf{r}_i + \alpha \text{sgn}(\nabla_{\mathbf{I}+\mathbf{r}_i}\mathcal{L})). \tag{2}$$

To facilitate an equitable comparison, the confidence scores undergo normalization within the range $[0, 1]$ by using the sigmoid function, which mitigates sensitivity to unbounded logit ranges (Wu et al., 2021). Maximizing $\mathcal{L}_{untargeted}$ can be achieved by making every target incorrectly predicted. For targeted attacks, we instead specify an adversarial label $l_i' \neq l_i$ for each target and minimize the following objective:

$$\mathcal{L}_{targeted} = \frac{1}{N} \sum_{i=1}^{N} [f_{cls}^{l_i'}(\mathbf{I} + \mathbf{r}, t_i) - f_{cls}^{l_i}(\mathbf{I} + \mathbf{r}, t_i)]. \tag{3}$$

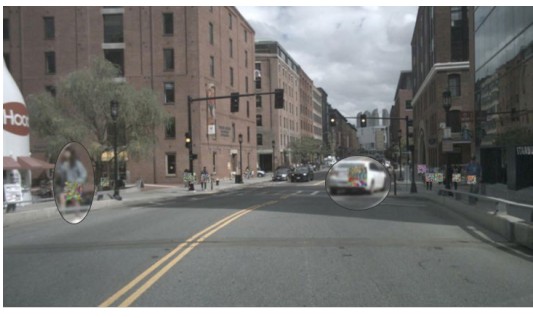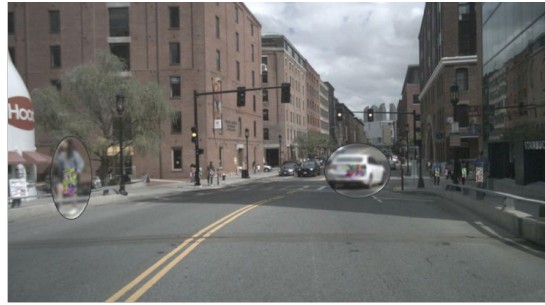

Figure 2: Illustration of adversarial patch size adaptations, wherein the patch size is adjusted proportionally to the target's 2D bounding box dimensions. The left panel depicts a fixed-size patch, while the right panel presents a dynamically scaled patch.

To attack the localization and other attributes, we adopt the straightforward $\mathcal{L}_1$ loss as the objective function, finding this method adequately effective:

$$\mathcal{L}_{localization} = \frac{1}{N} \sum_{i=1}^{N} ||f_{loc}(\mathbf{I} + \mathbf{r}, t_i) - loc_i||_1 + ||f_{orie}(\mathbf{I} + \mathbf{r}, t_i) - orie_i||_1 + ||f_{vel}(\mathbf{I} + \mathbf{r}, t_i) - vel_i||_1. \quad (4)$$

We further enhance our analysis by incorporating FGSM (Goodfellow et al., 2014), C&W Attack (Carlini & Wagner, 2017), and a stronger attacking method, AutoAttack (Croce & Hein, 2020). AutoAttack was originally designed for image classification tasks and can't be applied to object detection tasks directly. Therefore, we employ AutoPGD, a component of AutoAttack, as a more potent attack strategy. Our implementation rigorously adheres to the configurations specified in the foundational AutoAttack paper. The enhancements we introduce to the AutoPGD attack compared with the original PGD-Adv include the following modifications: (a) Integration of momentum during the update process, with the momentum. (b) Introduction of a dynamic step size that adjusts in accordance with the optimization process. (c) Implementation of a restart mechanism from the most effective attack points. (d) Balancing exploration and exploitation through the utilization of checkpoints.

Note that implementing pixel-based attacks in real-world scenarios is challenging because it requires altering camera-captured images in real-time. However, considering this type of attack is still crucial, particularly when attackers possess full knowledge of the model and can engage with real-time systems. Moreover, we can glean insights into the robustness of 3D object detectors in this adversarial setting.

### 4.2 Patch-based Attack

Following (Liu et al., 2018; Rossolini et al., 2022), we next turn our attention to patch-based adversarial attacks. Considering a target within a 3D bounding box, it can be characterized by its eight vertices and a central point, collectively denoted as $\{c_o, c_1, ..., c_8\}$ with $c_i \in \mathbb{R}^3$. Leveraging the camera parameters, we project these 3D points to 2D points on the image plane, yielding the transformed set $\{c'_o, c'_1, ..., c'_8\}$. We set the size of the adversarial patch to be proportional to the size of the rectangle formed by these 2D points, and strategically position it to be centered at point $c'_o$, as illustrated in Fig. 2. Note that the adversarial loss objectives for the patch-based attack remain consistent with those detailed in Sec. 4.1.

### 4.3 Black-box Attack

Building on the concept of universal adversarial perturbations in classification tasks, as extensively explored by (Moosavi-Dezfooli et al., 2017), we delve into the potential existence of universal adversarial patches specific to 3D object detection tasks. We start by defining and randomly initializing a fixed-size patch, which is then superimposed at the center of the object, as described in Sec. 4.2. We employ bilinear interpolation to resize this patch. During the training phase, we optimize the universal patch over a wide range of images using the Adam optimizer (Kingma & Ba, 2014), following the recommendations of (Moosavi-Dezfooli et al., 2017). In the testing phase, we apply the generated patch to unseen images and evaluate its performance

Table 2: Overall results: Clean results are evaluated on nuScenes validation set, and adversarial results are evaluated on the mini subset. The adversarial NDS is averaged for all the attack types and severities. †: trained with CBGS Zhu et al. (2019), §: re-trained models with minimal modification since there is no publicly available checkpoint. **BEV**: BEV-based representations. **Depth**: Explicit depth estimation.

| Models | Image Size | #param | BEV | Depth | Temporal | clean NDS | Adv NDS | clean mAP | Adv mAP |
|---|---|---|---|---|---|---|---|---|---|
| BEVFormer-Smal | $1280 \times 720$ | 59.6M | ✓ | | | 0.2623 | 0.1315 | 0.1324 | 0.0567 |
| BEVFormer-Base | $1600 \times 900$ | 69.1M | ✓ | | | 0.4128 | 0.1585 | 0.3461 | 0.0833 |
| DETR3D | $1600 \times 900$ | 53.8M | ✓ | | | 0.4223 | 0.1758 | 0.3469 | 0.1081 |
| DETR3D† | $1600 \times 900$ | 53.8M | ✓ | | | 0.4342 | 0.1953 | 0.3494 | 0.1126 |
| PETR-R50 | $1408 \times 512$ | 38.1M | ✓ | | | 0.3667 | 0.1193 | 0.3174 | 0.0641 |
| PETR-VovNet | $1600 \times 640$ | 83.1M | ✓ | | | 0.4550 | 0.1529 | 0.4035 | 0.0838 |
| BEVDepth-R50† | $704 \times 257$ | 53.1M | ✓ | ✓ | | 0.4057 | 0.1493 | 0.3327 | 0.0923 |
| BEVDepth-R101†§ | $704 \times 257$ | 72.1M | ✓ | ✓ | | 0.4167 | 0.1533 | 0.3376 | 0.1007 |
| BEVDet-R50† | $704 \times 257$ | 48.2M | ✓ | ✓ | | 0.3770 | 0.1069 | 0.2987 | 0.0634 |
| BEVDet-R101†§ | $704 \times 257$ | 67.2M | ✓ | ✓ | | 0.3864 | 0.1267 | 0.3021 | 0.0754 |
| BEVDet-Swin-Tiny† | $704 \times 257$ | 55.9M | ✓ | ✓ | | 0.4037 | 0.1074 | 0.3080 | 0.0635 |
| FCOS3D | $1600 \times 900$ | 55.1M | | - | | 0.3949 | 0.1339 | 0.3214 | 0.0714 |
| PGD-Det | $1600 \times 900$ | 56.2M | | - | | 0.4089 | 0.1441 | 0.3360 | 0.0843 |
| BEVFormer-Small | $1280 \times 720$ | 59.6M | ✓ | | ✓ | 0.4786 | 0.1593 | 0.3699 | 0.1007 |
| BEVFormer-Base | $1600 \times 900$ | 69.1M | ✓ | | ✓ | **0.5176** | 0.1445 | **0.4167** | 0.0846 |
| BEVDepth4D-R50† | $704 \times 257$ | 53.4M | ✓ | ✓ | ✓ | 0.4844 | **0.2144** | 0.3609 | **0.1211** |
| BEVDet4D-R50† | $704 \times 257$ | 48.2M | ✓ | ✓ | ✓ | 0.4570 | 0.1586 | 0.3215 | 0.0770 |

across various network architectures. The overall training pipeline for this approach is presented in Fig. 4(b), which simulates scenarios where attackers operate without detailed model knowledge or system access.

## 5 Experiments

### 5.1 Experimental Setup

To thoroughly assess the model performance, we evaluate both the clean performance and adversarial robustness using the nuScenes dataset. Given the substantial computational resources required for a full dataset evaluation, we opt for the nuScenes-mini dataset when probing adversarial robustness. We report two metrics, Mean Average Precision (mAP) and nuScenes Detection Score (NDS) (Caesar et al., 2020), in our experiments and discussions. For candidate models, wherever feasible, we use the official model configurations and publicly available checkpoints provided by open-sourced repositories; furthermore, we also train additional models with minimal modifications to facilitate experiments in controlled environments.

A holistic robustness evaluation necessitates examining models across varying degrees of attack intensity. Consequently, we introduce multiple severity levels for each attack — by increasing the iteration count for pixel-based attacks or by adjusting the size of the adversarial patch in patch-based attacks. For the detailed performance across these attack severities, interested readers are directed to the Appendix. Detailed parameter configurations for each type of attack are provided below:

**Pixel-based Attacks**. We evaluate pixel-based adversarial attacks using perturbations under the $\mathcal{L}_\infty$ norm. Our experiment setup fixes the maximum perturbation value at $\epsilon = 5$ and the step size at $\alpha = 0.1$. The process begins with the introduction of Gaussian noise to randomly perturb input images. Subsequently, we progressively increase the number of attack iterations, ranging from 1 to 50, for both untargeted and localization attacks. The iteration halts if no prediction results align with the ground truth. For localization attacks, we adjust the adversarial objective to $\mathcal{L}_1$ loss of the localization, orientation, and velocity predictions while keeping all other settings unchanged. Given that the nuScenes dataset contains six images for every scene with minimal overlap, our attacks targeted individual cameras. For the AutoPGD attack, we use an iteration of 10, the momentum is 0.75 and the initial step size is $0.2\epsilon$.

**Patch-based Attacks**. The initial patch pattern is generated using a Gaussian Distribution that has a mean and variance identical to the dataset. The attack iteration step size is designated as $\alpha = 5$ and we maintained the iteration number at 50. The patch scale is incrementally increased from 0.1 to 0.4.

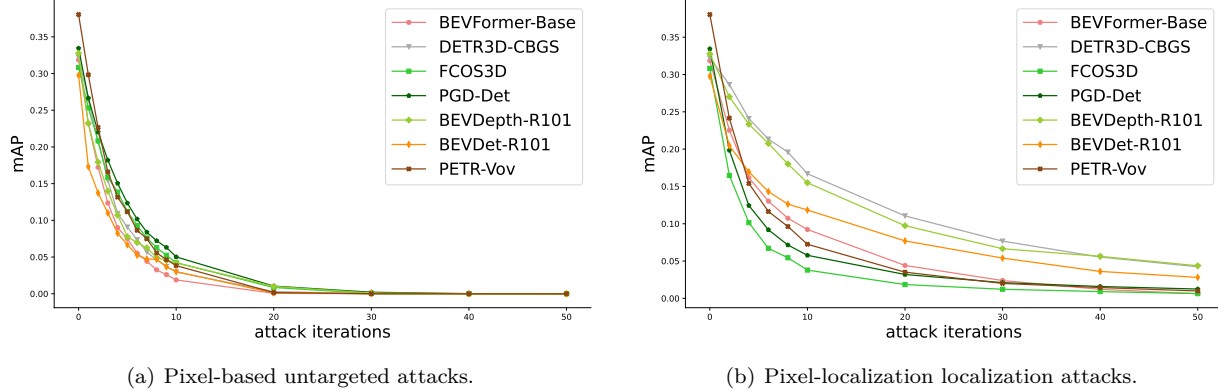

(a) Pixel-based untargeted attacks.

(b) Pixel-localization localization attacks.

Figure 3: Mean Average Precision (mAP) value *v.s* attack iterations. Models behave similarly under untargeted classification attacks while varying largely under localization attacks. All the models are similarly vulnerable to untargeted attacks while BEV-based exhibit better robustness toward localization attacks.

Table 3: Overall results: The adversarial NDS is averaged over all attack severities. †: trained with CBGS Zhu et al. (2019), §: re-trained models with minimal modification since there is no publicly available checkpoint. **BEV**: BEV-based representations. **Depth**: Explicit depth estimation. #: Using temporal modeling.

| Models | pix-cls | | pix-loc | | patch-cls | | patch-loc | | black-box | |
|---|---|---|---|---|---|---|---|---|---|---|
| | NDS | mAP | NDS | mAP | NDS | mAP | NDS | mAP | NDS | mAP |
| BEVFormer-Small | 0.1170 | 0.0284 | 0.1310 | 0.0836 | 0.1428 | 0.0425 | 0.1720 | 0.1096 | - | - |
| BEVFormer-Base | 0.1562 | 0.0621 | 0.1390 | 0.0892 | 0.1775 | 0.0713 | 0.1910 | 0.1560 | - | - |
| DETR3D | 0.1700 | 0.0796 | 0.1709 | 0.1454 | 0.1797 | 0.0664 | 0.2030 | 0.1663 | - | - |
| DETR3D† | 0.1921 | 0.0766 | 0.1873 | 0.1543 | 0.2021 | 0.0753 | 0.2183 | 0.1824 | 0.3523 | 0.2708 |
| PETR-R50 | 0.1256 | 0.0559 | 0.1170 | 0.0786 | 0.0887 | 0.0338 | 0.1330 | 0.0911 | 0.2350 | 0.1947 |
| PETR-VovNet† | 0.1708 | 0.0883 | 0.1321 | 0.0844 | 0.1352 | 0.0511 | 0.1548 | 0.0989 | - | - |
| BEVDepth-R50† | 0.1339 | 0.0646 | 0.1626 | 0.1301 | 0.1339 | 0.0603 | 0.1891 | 0.1366 | - | - |
| BEVDepth-R101†§ | 0.1436 | 0.0726 | 0.1691 | 0.1455 | 0.1301 | 0.0577 | 0.1751 | 0.1414 | 0.2801 | 0.1932 |
| BEVDet-R50† | 0.0806 | 0.0377 | 0.1244 | 0.0932 | 0.1102 | 0.0397 | 0.1562 | 0.1107 | - | - |
| BEVDet-R101†§ | 0.1121 | 0.0559 | 0.1406 | 0.1063 | 0.1176 | 0.0401 | 0.1558 | 0.1094 | 0.2113 | 0.1535 |
| BEVDet-Swin-Tiny† | 0.0856 | 0.0406 | 0.1058 | 0.0746 | 0.1365 | 0.0632 | 0.1580 | 0.1191 | - | - |
| FCOS3D | 0.1536 | 0.0861 | 0.1103 | 0.0524 | 0.1225 | 0.0543 | 0.1296 | 0.0799 | 0.2279 | 0.1830 |
| PGD-Det | 0.1696 | 0.0947 | 0.1105 | 0.0694 | 0.1126 | 0.0612 | 0.1621 | 0.1049 | 0.2437 | 0.1965 |
| BEVFormer-Small# | 0.1727 | 0.0964 | 0.1221 | 0.0949 | 0.1746 | 0.0907 | 0.1810 | 0.1388 | - | - |
| BEVFormer-Base# | 0.1328 | 0.0555 | 0.1312 | 0.1017 | 0.1689 | 0.0766 | 0.1910 | 0.1559 | 0.3018 | 0.2603 |
| BEVDepth4D-R50†# | 0.2143 | 0.0969 | 0.1914 | 0.1488 | 0.2388 | 0.0960 | 0.2425 | 0.1687 | - | - |
| BEVDet4D-R50†# | 0.1394 | 0.0473 | 0.1499 | 0.1031 | 0.1887 | 0.0633 | 0.2157 | 0.1358 | - | - |

**Black-box Attacks**. In this black-box setting, we optimize the patch with the nuScenes mini training set (Caesar et al., 2020). We set the learning rate to 10, the patch size to $100 \times 100$, and the patch scale to $s = 0.3$. Our adversarial objective uses targeted attacks, as in Eq. 3. The goal is to misclassify all categories as "Car" and to mislabel "Car" as "Pedestrian". In the inference stage, we apply the trained patch to unseen scenes and different models. To compensate for the occlusion effect induced by the patch, we set a baseline using a random pattern patch, and then assess the relative performance drop.

**Attacks for Temporal Models**. It is worth noting that BEVFormer incorporates historical features for temporal cross-attention. This suggests that attacks on prior frames can affect current predictions. Thereby, we simulate scenarios where attackers continuously attack multiple frames by modifying every single frame. On the other hand, for BEVDepth4D and BEVDet4D, we attack the current frame while leaving the history frames untouched. Such a setting can let us probe if benign temporal information can help in mitigating adversarial effects. More discussions can be found in Sec. 5.3.3.

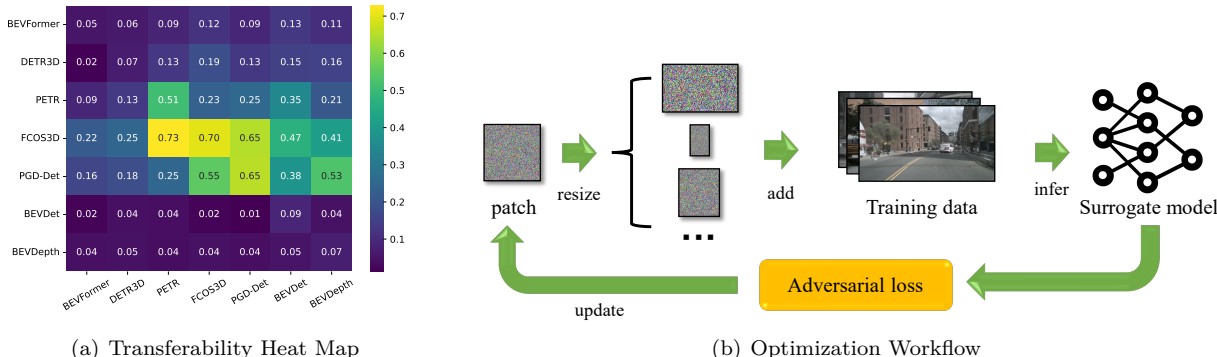

(a) Transferability Heat Map       (b) Optimization Workflow

Figure 4: Left panel: The horizontal axis corresponds to the targeted model while the vertical axis denotes the source model. Transferability is quantified by the proportional reduction in performance (specifically, mAP) in comparison to a randomized patch pattern of identical size. Right panel: The pipeline of the optimization process for the universal patch.

## 5.2 Main Results

Our main results of PGDAdv Attacks are presented in Tab. 2 and Tab. 3. The Adv NDS and mAP are averaged across all attack types and severities, excluding only the black-box attacks. The results of the AutoPGD Attack are present in Tab. 4. The results of FGSM and C&W attacks can be found in Appendix A.

Table 4: Results of AutoPGD Croce & Hein (2020) attack.

| Model | NDS | mAP | mATE | mASE | mAOE | mAVE | mAAE |
|---|---|---|---|---|---|---|---|
| DETR3D | 0.1373 | 0.0522 | 0.9419 | 0.5150 | 1.0066 | 1.2340 | 0.4311 |
| PETR | 0.0880 | 0.0094 | 1.0032 | 0.6438 | 0.8749 | 1.2613 | 0.6482 |
| FCOS3D | 0.1562 | 0.0507 | 0.8930 | 0.4965 | 0.9639 | 1.0070 | 0.3380 |
| PGD-Dev | 0.1700 | 0.0544 | 0.8494 | 0.5297 | 0.8157 | 1.2942 | 0.3775 |
| BEVDet | 0.0766 | 0.0203 | 1.0156 | 0.6403 | 1.0308 | 1.1250 | 0.6847 |

Overall, we interestingly note that all the existing camera-based detectors are vulnerable to adversarial attacks, *e.g.*, the adversarial NDS of all models (except BEVDepth-4D) is lower than 0.2 for PGD-Adv Attacks. Furthermore, we find that AutoPGD can severely compromise the performance of the detection models by only leveraging 10 steps of iteration, as shown in Tab. 4. In terms of the attack categories, pixel-based attacks tend to be more malicious than patch-based ones, indicating that superimposing patch patterns onto target objects generally leads to less adversarial effects than pixel alterations. This finding concurs with the understanding that adversarial patches, being modifications of only specific image segments, naturally cause restricted adversarial perturbations. Furthermore, we find that attacks that aim to confuse classification have a greater adversarial effect than those meant to mislead localization. This observation holds for both pixel-based and patch-based attacks. Nevertheless, as illustrated in Fig. 3, the discrepancy in model robustness is more pronounced under localization attacks, indicating a variable degree of vulnerability in accurately identifying object locations.

For black-box attacks, adversarial examples from monocular detectors exhibit enhanced transferability, even to BEV-based models, as illustrated in Fig. 4(a). Universal patches trained using FCOS3D or PGD-Det demonstrate strong transferability among various models. For instance, transferring attacks from FCOS3D to PETR led to a relative performance decline exceeding 70%. On the other hand, BEVDet and BEVDepth produce adversarial examples with limited transferability and show reduced susceptibility to universal patch attacks. Notably, the patch maintains its adversarial nature even after resizing to different shapes and scales. Such a universal patch, spanning various images, models, and scales, presents a pronounced potential threat for camera-based detectors, especially given the zero-risk tolerance in autonomous driving contexts.

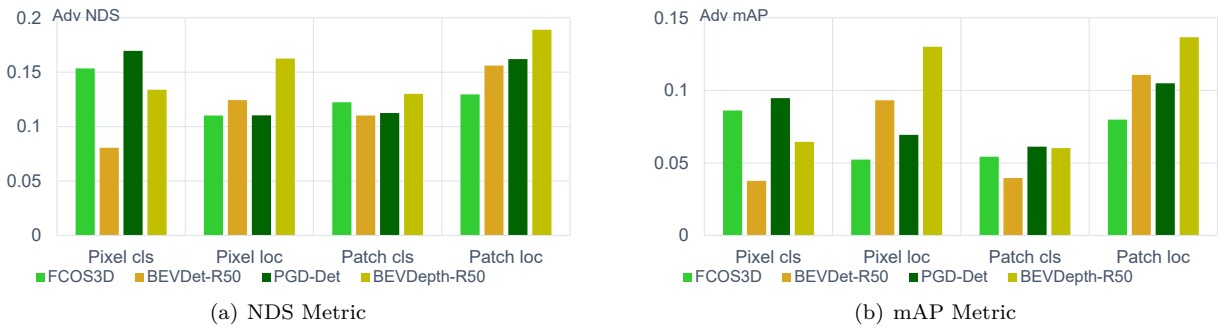

(a) NDS Metric

(b) mAP Metric

Figure 5: Comparisons between BEV-based models and non-BEV-based models.

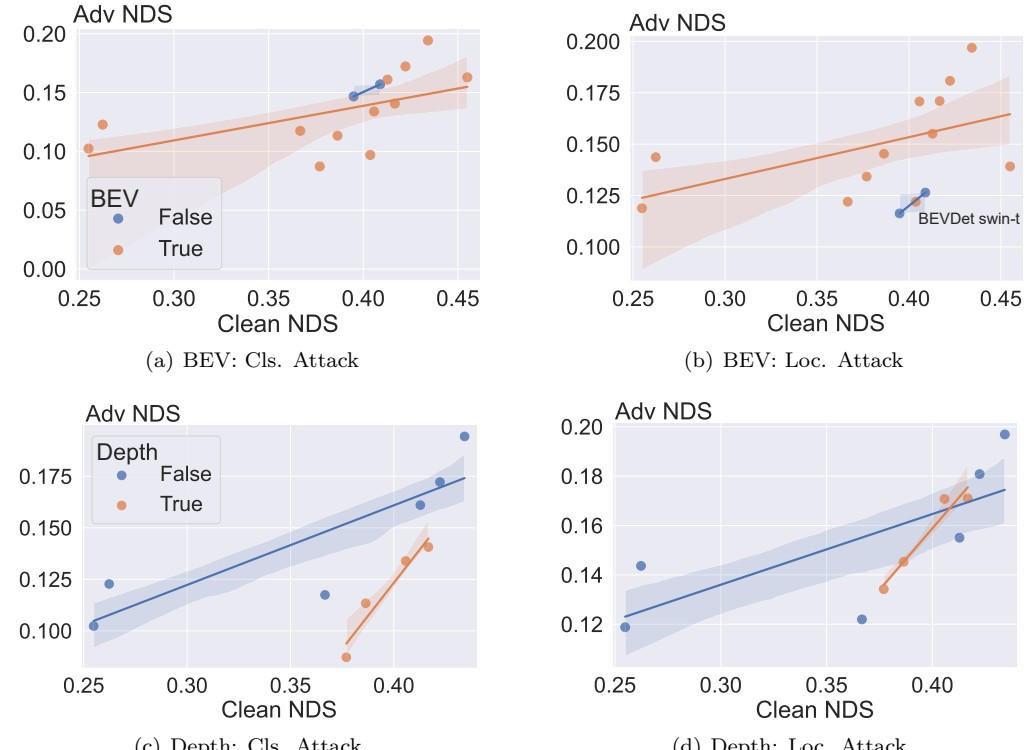

(a) BEV: Cls. Attack

(b) BEV: Loc. Attack

(c) Depth: Cls. Attack

(d) Depth: Loc. Attack

Figure 6: Left: Comparison between non-BEV-based models and BEV-based models. Right: Comparison between depth-based and depth-free models.

## 5.3 Discussions

We next provide an in-depth discussion about these results. For better analysis, we primarily focus on three model components: *BEV representation*, *Depth* (*i.e.*, the incorporation of an explicit depth estimation branch for BEV transformation), and *Temporal Modeling* (*i.e.*, the ability to learn from multi-frame inputs).

### 5.3.1 BEV-based Representations

***BEV-based models are generally vulnerable to classification attacks but show notable robustness to localization attacks.*** To probe if BEV detectors retain their superiority over monocular detectors under adversarial attacks, we select four models for comparison: BEVDet (Huang et al., 2021), BEVDepth (Li et al., 2022a), FCOS3D (Wang et al., 2021), and PGD-Det (Wang et al., 2022a), as they exhibit similar performance under standard conditions and all employ the ResNet101 backbone. As shown in Fig. 5, we can observe that BEV-based methods generally fail to showcase a clear advantage in terms of robustness

against untargeted attacks. However, we interestingly note that BEV-based models demonstrate superior performance under localization attacks — under the pixel-based localization attacks, the adversarial NDS of BEVDepth outperforms PGD-Det by about 53%. This conclusion is further corroborated in Fig. 6(a) and Fig. 6(b).

### 5.3.2 Explicit Depth Estimation

***Precise depth estimation is crucial for depth-based models. Additionally, depth-free methods have the potential to yield stronger robustness.*** Our analysis suggests that models with more precise depth estimation capabilities typically exhibit enhanced robustness against adversarial attacks. As seen in Fig. 5, PGD-Det outperforms FCOS3D by leveraging superior depth estimation. This enhanced depth estimation results in consistent robustness improvement across all attack types. Additionally, the comparison between BEVDet and BEVDepth, which differ only in their depth estimation module, shows that the accurate depth estimation in BEVDepth can lead to a 39.6% increase in robustness.

Furthermore, we found that depth-estimation-free approaches (Li et al., 2022b; Wang et al., 2022b; Liu et al., 2022a) generally show advantages over depth-based detectors under classification attacks (Fig. 6(c)). Interestingly, for localization attacks, depth-based models can outperform some depth-free models, if the depth estimation is sufficiently accurate, as shown in Fig. 6(d). Nonetheless, DETR3D (Wang et al., 2022b) still shows the best robustness, suggesting that carefully designed depth-free methods have the potential for superior robustness.

### 5.3.3 Temporal Fusion

***The effects of adversarial attacks can be mitigated using clean temporal information, but they might be exacerbated when multi-frame adversarial inputs are used.*** To investigate the impact of temporal information on adversarial robustness, we introduce two distinct attack scenarios. For the BEVFormer model, which updates its history of BEV queries on the fly, we attack each input frame. This results in all sequential inputs used for temporal information modeling being adversarial examples, causing an accumulation of errors within the model through retained temporal data. Our experiments reveal that the BEVFormer-Base model, when using temporal information, underperforms compared to its single-frame variant (*i.e.*, 0.1585 *v.s.* 0.1445). To further demonstrate the influence of temporal fusion, we simulate three cases: (a) Benign case: The model processes clean input across multiple frames. (b) Continuous adversarial attack: The model processes adversarial input persistently across multiple frames. (c) Single adversarial attack: The model processes clean input followed by adversarial input at a singular frame. We use the benign case as the ground truth and calculate the error on BEV temporal features according to it. The results presented in Tab. 5 further prove the observation. In the second scenario, we attack the current timestamp input while preserving historical information untouched. Under this condition, we test BEVDepth4D and BEVDet4D, which integrate features from the current frame and a recent historical frame in their predictions. As shown in Tab. 2, we observe that clean temporal information significantly reduces the adversarial effect in this scenario, with 0.1493 *v.s.* 0.2144 for BEVDepth, and 0.1069 *v.s.* 0.1586 for BEVDet.

Table 5: Error between adversarial BEV features and benign BEV features under different frames.

| Scenario | frame1 | frame3 | frame5 | frame7 | frame9 | frame11 |
|---|---|---|---|---|---|---|
| Continuous adversarial attack | 0.11 | 0.27 | 0.23 | 0.28 | 0.25 | 0.22 |
| Single adversarial attack | 0 | 0 | 0 | 0 | 0 | 0.16 |

### 5.3.4 Others

***Strategies specifically designed to tackle long-tail problems can concurrently improve model robustness.*** Object detection often faces long-tail challenges, where certain categories like "Car" and "Pedestrian" are significantly more prevalent than others, such as "Motorcycle". Our findings suggest that

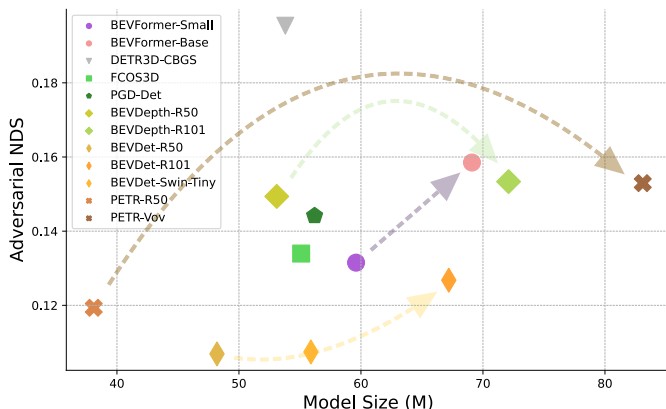

Figure 7: Comparison between different model sizes: For the same model, an increase in parameter size typically leads to enhanced robustness against adversarial attacks.

strategies designed to address long-tail problems, such as class-balanced group sampling (CBGS) training (Zhu et al., 2019), can also improve robustness. Our results show that DETR3D trained with CBGS improves adversarial NDS by about 11%. This observation contrasts with (Wu et al., 2021), which suggested that resampling training strategies minimally impact robustness. Nonetheless, it is important to note that our study considers detection tasks, which are different from the classification tasks in (Wu et al., 2021).

***Increasing the backbone size consistently leads to improved robustness.*** We investigate the impact of different backbone architectures, including ResNet (He et al., 2016), VoVNet (Lee et al., 2019), and Swin-Transformer (Liu et al., 2021), on the robustness of models. We first compare Swin-Tiny and ResNet50, as their parameter sizes are similar (23.6M *v.s.* 27.5M). Although they perform slightly differently under various attacks, the overall robustness of ResNet50 and Swin-Tiny is similar (*i.e.*, 0.1069 *v.s.* 0.1074). On the other hand, the VovNet outperforms in both standard and adversarial robustness. Additionally, we find increasing the backbone size consistently leads to improved robustness. This trend is particularly noticeable for models with weaker robustness, as illustrated in Fig. 7.

## 6 Ethical and Societal Considerations

This study addresses the threats of adversarial attacks in camera-based 3D detection models, focusing primarily on digital attacks. We emphasize the need to extend this research to include physical attack scenarios in future work, due to their significant potential impact on autonomous driving systems. Ethically, our research highlights the crucial responsibility of developers to ensure the safety and reliability of these systems, especially in public spaces where they are more vulnerable to adversarial attacks. The broader impacts of our study underline the importance of integrating strong security measures in both the development and deployment of such technologies. This is vital to reduce the risk of harm to society and promote a safer, more ethical approach to using technology in critical safety applications.

## 7 Conclusions

We conduct an exhaustive analysis of adversarial robustness in camera-based 3D object detection models. Our findings reveal that a model's robustness does not necessarily align with its performance under normal conditions. Through our investigation, we successfully pinpoint several strategies that can enhance robustness. We hope our findings will contribute valuable insights to the development of more robust camera-based object detectors in the future.

## Acknowledgement

This work is partially supported by a gift from Open Philanthropy and UCSC Office of Research Seed Funding for Early Stage Initiatives. This work is based upon the work supported by the National Center for Transportation Cybersecurity and Resiliency (TraCR) (a U.S. Department of Transportation National University Transportation Center) headquartered at Clemson University, Clemson, South Carolina, USA. Any opinions, findings, conclusions, and recommendations expressed in this material are those of the author(s) and do not necessarily reflect the views of TraCR, and the U.S. Government assumes no liability for the contents or use thereof.

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

## A  More Adversarial Attacks Results

We provide the results of FGSM (Goodfellow et al., 2014) and C&W attack (Carlini & Wagner, 2017) in this section. The results can be found fron Tab. 6 to Tab. 9. We observe that in the context of 3D object detection, even single-step FGSM can compromise the performance of models to a large extent, which further reveals the vulnerability of these models. The attack effectiveness of the C&W attack is close to the FGSM attack. However, the C&W attack needs more steps to optimize, which might limit their real-world application usage.

Table 6: Results of FGSM Classification attack.

| Model | NDS | mAP | mATE | mASE | mAOE | mAVE | mAAE |
|--------|--------|--------|--------|--------|--------|--------|--------|
| DETR3D | 0.2259 | 0.1406 | 0.9066 | 0.4804 | 0.7946 | 0.9122 | 0.3498 |
| PETR | 0.1501 | 0.0710 | 0.9419 | 0.5616 | 0.9664 | 0.9048 | 0.4789 |
| BEVDet | 0.1427 | 0.0603 | 1.0925 | 0.4855 | 1.1231 | 1.1421 | 0.3884 |
| FCOS3D | 0.1712 | 0.1104 | 0.9319 | 0.5034 | 1.0274 | 1.3144 | 0.4053 |
| PGD-Dev | 0.1848 | 0.0865 | 0.9440 | 0.5292 | 0.8091 | 0.9069 | 0.3949 |

Table 7: Results of FGSM Localization attack.

| Model | NDS | mAP | mATE | mASE | mAOE | mAVE | mAAE |
|--------|--------|--------|--------|--------|--------|--------|--------|
| DETR3D | 0.2441 | 0.2095 | 0.8805 | 0.4892 | 0.7957 | 1.1887 | 0.4413 |
| PETR | 0.1778 | 0.1273 | 0.9900 | 0.4739 | 1.0308 | 1.1836 | 0.3951 |
| BEVDet | 0.1675 | 0.1249 | 0.9852 | 0.4984 | 1.0121 | 1.2842 | 0.4655 |
| FCOS3D | 0.1583 | 0.0942 | 0.9647 | 0.4982 | 0.9403 | 1.5566 | 0.4852 |
| PGD-Dev | 0.1606 | 0.0960 | 1.0325 | 0.5231 | 0.9541 | 1.0494 | 0.3963 |

Table 8: Results of C&W Classification attack.

| Model | NDS | mAP | mATE | mASE | mAOE | mAVE | mAAE |
|--------|--------|--------|--------|--------|--------|--------|--------|
| DETR3D | 0.2765 | 0.2030 | 0.8686 | 0.4752 | 0.7202 | 0.8435 | 0.3419 |
| PETR | 0.1656 | 0.1167 | 0.9507 | 0.4833 | 1.0228 | 1.5613 | 0.4938 |
| FCOS3D | 0.1119 | 0.0730 | 1.0815 | 0.5873 | 1.0157 | 1.0151 | 0.6586 |
| PGD-Dev | 0.1585 | 0.0924 | 1.0400 | 0.5003 | 0.8569 | 1.3132 | 0.5195 |

Table 9: Results of C&W Localization attack.

| Model | NDS | mAP | mATE | mASE | mAOE | mAVE | mAAE |
|--------|--------|--------|--------|--------|--------|--------|--------|
| DETR3D | 0.2793 | 0.1461 | 0.8809 | 0.4635 | 0.7118 | 0.5647 | 0.3165 |
| PETR | 0.1681 | 0.0870 | 0.8943 | 0.4938 | 0.9151 | 1.3880 | 0.4513 |
| FCOS3D | 0.1217 | 0.0632 | 1.0194 | 0.5774 | 0.8748 | 1.4137 | 0.6470 |
| PGD-Dev | 0.1485 | 0.0761 | 0.9929 | 0.5214 | 0.9067 | 1.3131 | 0.4745 |

## B  Dynamical Patch *v.s.* Fixed-size Patch

We compare the results of the dynamical patch and fixed-size patch. In our paper, we choose to use dynamical size patches because it is more physically reasonable. Real-world patch size changes according to distance from sensors. Attackers only need a smaller patch for pedestrians to fool the detectors while might need a

larger one for larger objects (*i.e.*, Car and Bus). As a result, it is not reasonable nor fair to apply a fixed patch size to every object. To explain the difference, we calculate the detection results of each class, the comparison can be seen in Fig. 8. The results are evaluated using BEVFormer-Base(Li et al., 2022b) with temporal information on nuScenes(Caesar et al., 2020) mini validation set. We calculate the relative AP drop compared to clean input.

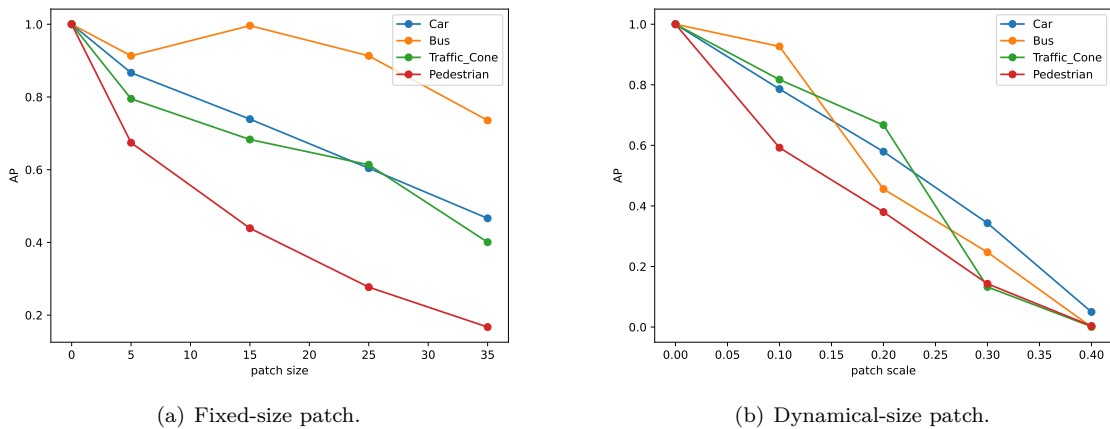

(a) Fixed-size patch.                    (b) Dynamical-size patch.

Figure 8: Comparison between different patch size settings.

Table 10: For PGD-Det model, the adversarial universal patch trained on nuScenes can transfer to KITTI.

| Type | Easy | Moderate | Hard |
|---|---|---|---|
| Clean | 64.4 | 54.5 | 49.2 |
| Random Patch | 53.6 | 44.4 | 39.4 |
| Adv Patch | 44.2 | 37.2 | 32.8 |

## C  Black-Box Transfer Attacks

Here we provide the full results of universal patch-based black box transfer attacks as shown in Tab. 11 and Tab. 12. We conduct universal patch attacks using BEVFormer-base(Li et al., 2022b) without temporal information, DETR3D(Wang et al., 2022b), PETR-R50(Liu et al., 2022a), BEVDepth-R101(Li et al., 2022a), and BEVDet-R50(Huang et al., 2021). Among the above models, DETR3D(Wang et al., 2022b), BEVDet(Huang et al., 2021), and BEVDepth(Li et al., 2022a) are trained with CBGS strategy. Considering the occlusion induced by patches, we randomly initialize the patch with the same size to serve as the baseline. To further verify the transferability of the generated universal patch, we utilize the universal patch trained by PGD-Det to KITTI (Geiger et al., 2012) dataset. We show that the universal patch generated using nuScenes (Caesar et al., 2020) can effectively transfer to KITTI (Geiger et al., 2012), as shown in Tab. 10. This further validates the existence of universal adversarial patterns in 3D detection tasks.

## D  Visualization of Depth Estimation

We visualize the depth estimation results of BEVDet (Huang et al., 2021) and BEVDepth (Li et al., 2022a) in Fig. 9 to further show the importance of precise depth estimation for depth-based approaches. Due to the guidance from sparse LiDAR points, BEVDepth consistently yielded superior depth estimation results, contributing to its heightened robustness.

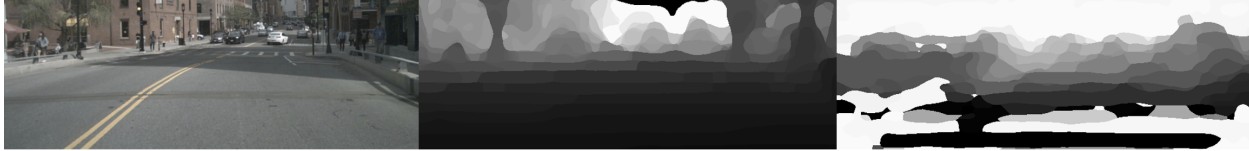

Figure 9: Depth estimation results. From left to right: original FRONT camera images, BEVDepth (Li et al., 2022a) depth prediction results, BEVDet (Huang et al., 2021) depth prediction results. Accurate depth estimation provides the model with strong robustness.

| Adversarial Source | BEVFormer | DETR3D | PETR | FCOS3D | PGD-Det | BEVDet | BEVDepth |
|---|---|---|---|---|---|---|---|
| Random Noise | 0.2816 | 0.3014 | 0.2515 | 0.2379 | 0.2546 | 0.1924 | 0.2380 |
| BEVFormer | 0.2663 | 0.2827 | 0.2301 | 0.2098 | 0.2316 | 0.1676 | 0.2132 |
| DETR3D | 0.2767 | 0.2799 | 0.2195 | 0.1936 | 0.2221 | 0.1634 | 0.2017 |
| PETR | 0.2561 | 0.2613 | 0.1236 | 0.1825 | 0.1904 | 0.1256 | 0.1895 |
| FCOS3D | 0.2205 | 0.2266 | 0.0683 | 0.0724 | 0.0893 | 0.1018 | 0.1410 |
| PGD-Det | 0.2357 | 0.2463 | 0.1805 | 0.1069 | 0.0879 | 0.1186 | 0.1128 |
| BEVDet | 0.2760 | 0.2807 | 0.2416 | 0.2334 | 0.2513 | 0.1758 | 0.2283 |
| BEVDepth | 0.2693 | 0.2874 | 0.2422 | 0.2275 | 0.2445 | 0.1829 | 0.2213 |

Table 11: Universal patch black box attacks: full results of mAP. The X-axis represents the target models and Y-axis represents the source white box models.

| Adversarial Source | BEVFormer | DETR3D | PETR | FCOS3D | PGD-Det | BEVDet | BEVDepth |
|---|---|---|---|---|---|---|---|
| Random Noise | 0.3179 | 0.3746 | 0.2857 | 0.2776 | 0.2919 | 0.2670 | 0.3206 |
| BEVFormer | 0.3141 | 0.3589 | 0.2648 | 0.2475 | 0.2696 | 0.2254 | 0.2965 |
| DETR3D | 0.3235 | 0.3609 | 0.2566 | 0.2264 | 0.2592 | 0.2177 | 0.2876 |
| PETR | 0.2954 | 0.3412 | 0.1890 | 0.2127 | 0.2396 | 0.1769 | 0.2804 |
| FCOS3D | 0.2604 | 0.3149 | 0.1125 | 0.1473 | 0.1669 | 0.1461 | 0.2336 |
| PGD-Det | 0.2811 | 0.3350 | 0.2315 | 0.1733 | 0.1670 | 0.1552 | 0.2080 |
| BEVDet | 0.3079 | 0.3672 | 0.2715 | 0.2734 | 0.2807 | 0.2551 | 0.3131 |
| BEVDepth | 0.3140 | 0.3655 | 0.2687 | 0.2648 | 0.2750 | 0.2467 | 0.3011 |

Table 12: Universal patch black box attacks: full results of NDS. The X-axis represents the target models and Y-axis represents the source white box models.

# E Full Results

In this part, we list the full experiment results. We present the mAP and NDS metrics of all the models under 4 types of attacks (*i.e.*, the attack iterations or the patch scale). For each attack, we use multiple attack severities to minimize randomness. We set the random seed to 0 for all experiments. The curves are plotted from Fig. 10 to Fig. 13.

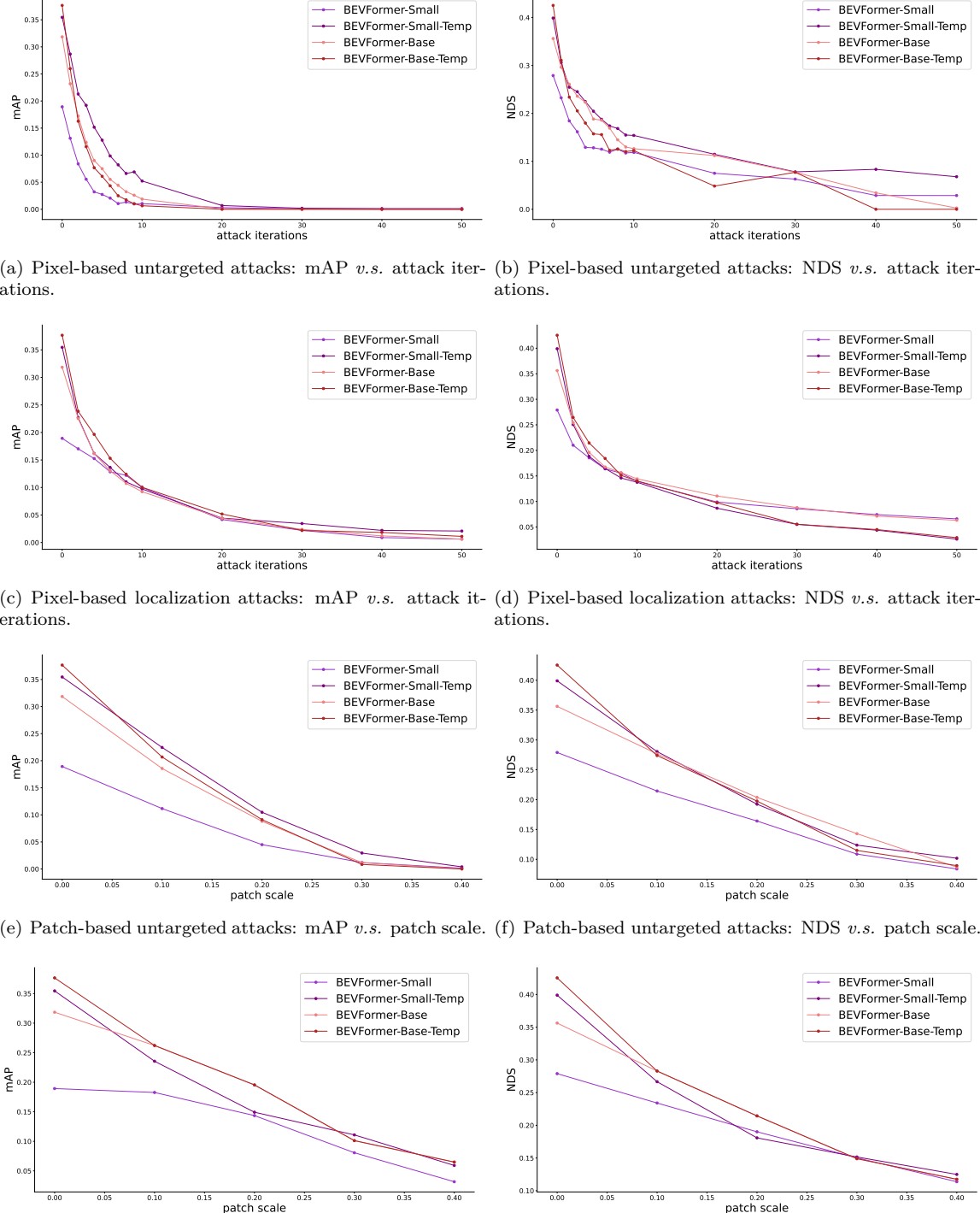

(a) Pixel-based untargeted attacks: mAP *v.s.* attack iterations.

(b) Pixel-based untargeted attacks: NDS *v.s.* attack iterations.

(c) Pixel-based localization attacks: mAP *v.s.* attack iterations.

(d) Pixel-based localization attacks: NDS *v.s.* attack iterations.

(e) Patch-based untargeted attacks: mAP *v.s.* patch scale.

(f) Patch-based untargeted attacks: NDS *v.s.* patch scale.

(g) Patch-based localization attacks: mAP *v.s.* patch scale.

(h) Patch-based localization attacks: NDS *v.s.* patch scale.

Figure 10: BEVFormer full results.

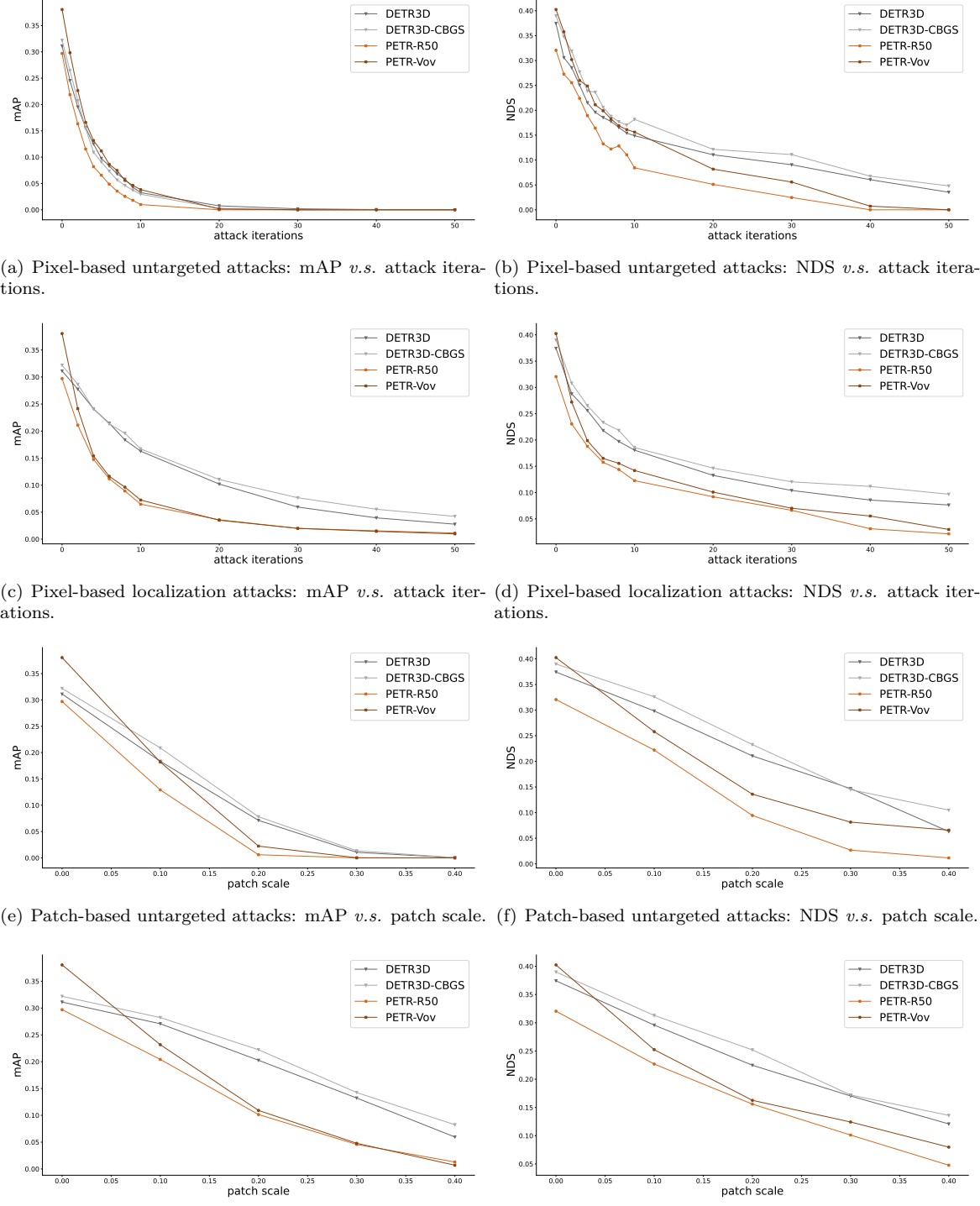

(a) Pixel-based untargeted attacks: mAP *v.s.* attack iterations.

(b) Pixel-based untargeted attacks: NDS *v.s.* attack iterations.

(c) Pixel-based localization attacks: mAP *v.s.* attack iterations.

(d) Pixel-based localization attacks: NDS *v.s.* attack iterations.

(e) Patch-based untargeted attacks: mAP *v.s.* patch scale.

(f) Patch-based untargeted attacks: NDS *v.s.* patch scale.

(g) Patch-based localization attacks: mAP *v.s.* patch scale.

(h) Patch-based localization attacks: NDS *v.s.* patch scale.

Figure 11: DETR3D and PETR full results.

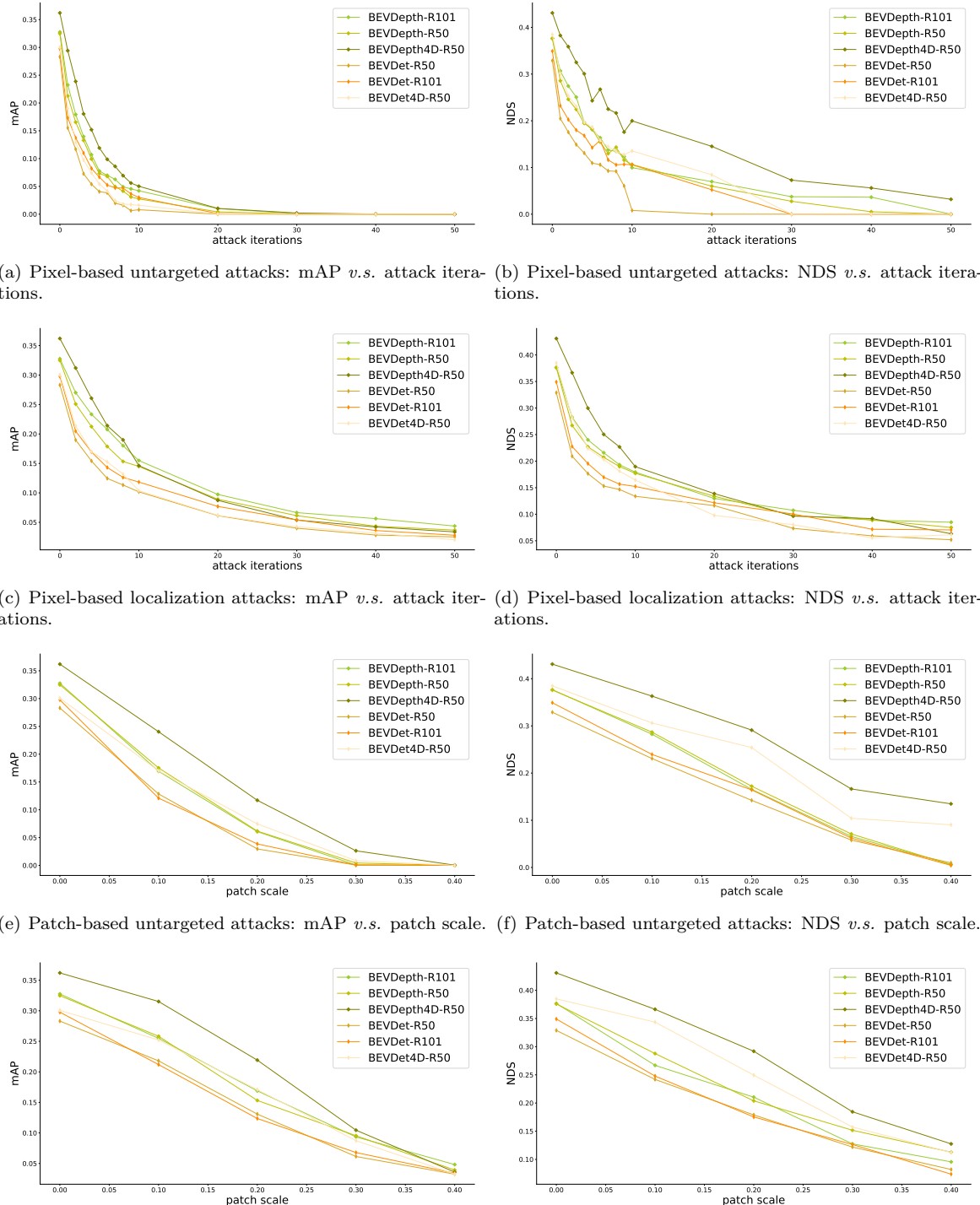

(a) Pixel-based untargeted attacks: mAP *v.s.* attack iterations.

(b) Pixel-based untargeted attacks: NDS *v.s.* attack iterations.

(c) Pixel-based localization attacks: mAP *v.s.* attack iterations.

(d) Pixel-based localization attacks: NDS *v.s.* attack iterations.

(e) Patch-based untargeted attacks: mAP *v.s.* patch scale.

(f) Patch-based untargeted attacks: NDS *v.s.* patch scale.

(g) Patch-based localization attacks: mAP *v.s.* patch scale.

(h) Patch-based localization attacks: NDS *v.s.* patch scale.

Figure 12: BEVDet and BEVDepth full results.

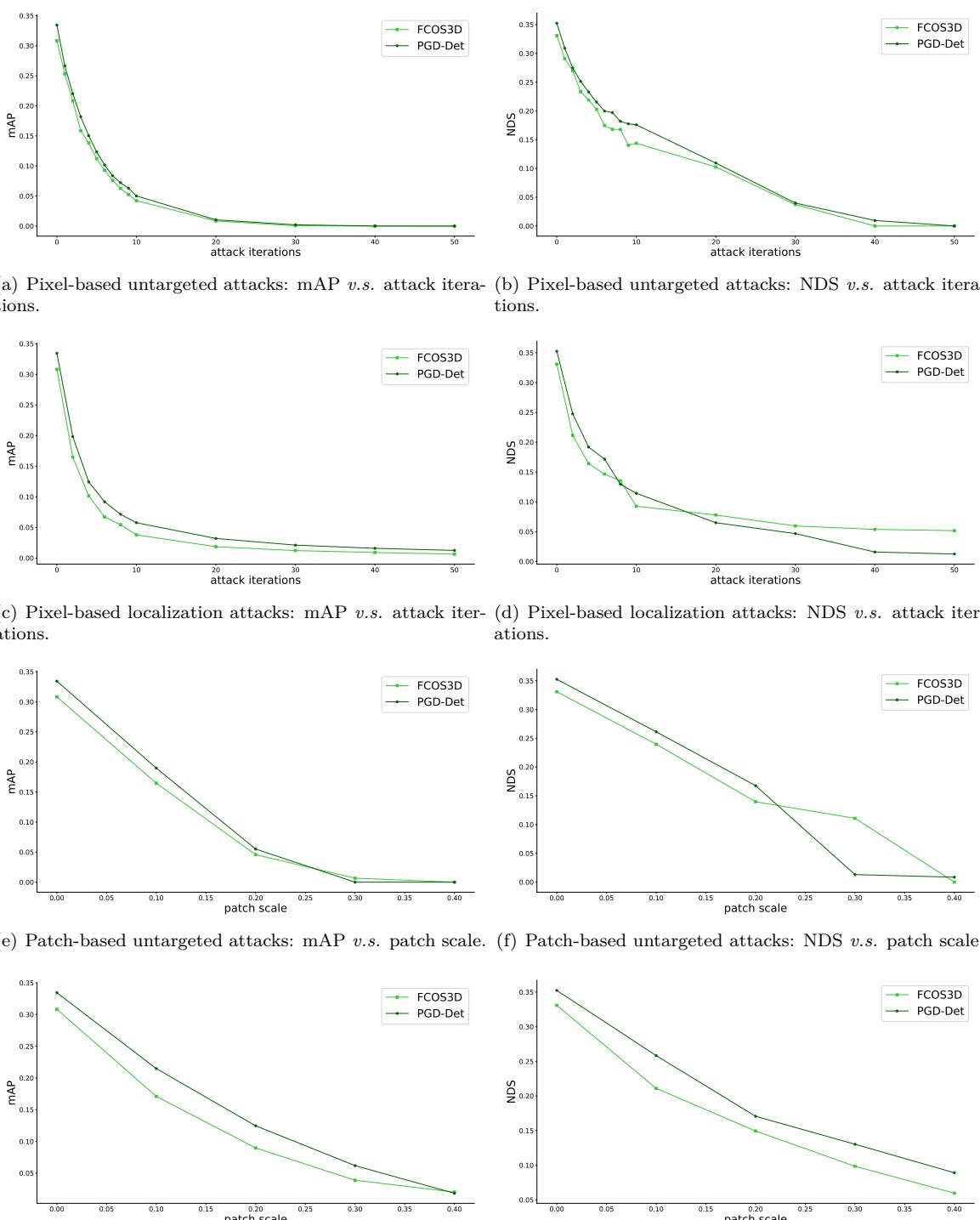

(a) Pixel-based untargeted attacks: mAP *v.s.* attack iterations.

(b) Pixel-based untargeted attacks: NDS *v.s.* attack iterations.

(c) Pixel-based localization attacks: mAP *v.s.* attack iterations.

(d) Pixel-based localization attacks: NDS *v.s.* attack iterations.

(e) Patch-based untargeted attacks: mAP *v.s.* patch scale.

(f) Patch-based untargeted attacks: NDS *v.s.* patch scale.

(g) Patch-based localization attacks: mAP *v.s.* patch scale.

(h) Patch-based localization attacks: NDS *v.s.* patch scale.

Figure 13: FCOS3D and PGD-Det full results

