# OpenReview forum: "On the Adversarial Robustness of Camera-based 3D Object Detection"
_TMLR — Accepted by TMLR_

### Review · Reviewer_EhUz · 2023-10-31

**Summary Of Contributions:**

This paper conducts an investigation of the robustness of leading camera-based 3D object detection approaches under various adversarial conditions. It evaluates the robustness against pixel-based and patch-based adversarial attacks in classification and localization tasks. The paper found four techniques can potentially help improve the robustness: bird's-eye-view-based representations, depth-estimation-free approaches, accurate depth estimation, and multi-frame benign inputs. The authors conduct comprehensive experiments and show interesting results on designs that can benefit the adversarial robustness.

**Audience:**

Yes

**Broader Impact Concerns:**

No concern.

**Claims And Evidence:**

Yes

**Requested Changes:**

See weakness above.

**Strengths And Weaknesses:**

**Strengths**

1. The experimental setting and analysis is exhaustive, providing insightful indications on robust 3D model design.
2. The figures and the tables are self-explanatory. The paper is well-written and is easy to follow.

**Weakness**

1. The paper shows several techniques (i.e., BEV-based representation, explicit depth estimation, temporal fusion, etc.) can help improve the adversarial robustness by comparing the empirical adversarial robustness of models using different combinations of these techniques. It would be better if the authors could provide deeper understanding on this and explain why these techniques could help using theoretical analysis or a more fine-grained ablation study.
2. I am curious what is the strongest attack for 3D models. The authors considered PGD-based methods and patch-based methods, which is great. But I would like to learn how is the model's performance against even stronger attack like AutoAttack (Croce et al., 2020).

---

> ### Author Response · Authors · 2023-11-22
> **Re: Review of Paper1724 by Reviewer EhUz**
>
> Thank you for your insightful feedback on our paper. We appreciate the opportunity to clarify and discuss the points raised in your comments.
>
> >  ***Question 1**: It would be better if the authors could provide a deeper understanding of this and explain why these techniques could help using theoretical analysis or a more fine-grained ablation study.*
>
> **Answer 1**: We thank you for highlighting this aspect of our study. We concur with the necessity for a more in-depth analysis of our findings. Accordingly, we have undertaken examinations of the temporal fusion's impact and the explicit depth estimation in the following manner:
>
> **Temporal:** To investigate the effects of temporal fusion, we utilized BEVFormer, which integrates temporal information dynamically and has demonstrated notable benefits from such fusion. Our experimental framework covers three scenarios:
>
> 1. Benign Scenario: The model processes clean input across multiple frames.
> 2. Continuous Adversarial Scenario: The model processes adversarial input persistently across multiple frames.
> 3. Single Adversarial Scenario: The model processes clean input followed by adversarial input at a singular frame.
>
> We use the benign case as ground truth and calculate the error on temporal features according to it. The results are shown below:
>
> |                                 | frame1 | frame3 | frame5 | frame7 | frame9 | frame11 |
> | ------------------------------- | ------ | ------ | ------ | ------ | ------ | ------- |
> | Continuous Adversarial Scenario | 0.11   | 0.27   | 0.23   | 0.28   | 0.25   | 0.22    |
> | Single Adversarial Scenario     | 0      | 0      | 0      | 0      | 0      | 0.16    |
>
> The results indicate that prolonged exposure to adversarial examples leads to an incremental error accumulation, leading to greater discrepancies in subsequent frames. However, the adversarial interference to a single frame allows benign temporal data to mitigate the adversarial impact, as shown by the reduced error at the 11th frame. This observation corroborates our empirical findings.
>
> **Depth Estimation**: To study the influence of explicit depth estimation, we separate the depth estimation and feature extraction branches within BEVDet. We computed the relative error for both the depth map and the feature map. We calculate the relative error of each map, the depth estimation map, and the feature map. The formula is $err = abs((clean - adv)/(clean + eps))$. The PGD attack was used to disrupt localization predictions, with the results shown below:
>
> | Feature | BEV Feature | Image Feature | Depth Map |
> | ------- | ----------- | ------------- | --------- |
> | Error   | 23004.18    | 7.15          | 4714.67   |
>
> The BEV feature, derived from the image feature and directly linked to depth, shows an error reflective of the multiplicative effect between image features and depth predictions. Notably, the depth prediction substantially contributes to the error within the BEV feature, showing that for depth-dependent BEV models, erroneous depth estimations under adversarial conditions can significantly distort the final BEV representation, thereby degrading model performance. Conversely, models without explicit depth estimation don’t suffer from inaccurate depth estimation.
>
> **BEV Feature:** We recognize the limitations inherent in executing fine-grained experiments to study the mechanisms by which BEV representation enhances localization attack robustness. The distinct divergence between BEV-based and non-BEV-based representations presents inherent challenges in isolating BEV representation for a comparative ablation study. Nevertheless, the empirical evidence presented in our manuscript—shown through Figure 6 (b) solidly demonstrates that BEV representation inherently possesses greater robustness to localization attacks. A comprehensive investigation into the intrinsic factors contributing to this robustness remains a prospective avenue for future research.

---

> ### Author Response · Authors · 2023-11-22
> **Re: Review of Paper1724 by Reviewer EhUz**
>
> >  ***Question 2**: How is the model's performance against an even stronger attack?*
>
> **Answer 2**: Thanks for your suggestions. It’s important to note that our work mainly focuses on how the model design will behave differently under attacks to study their adversarial robustness. Therefore, using strong attacks to fully crash the models’ performance isn’t our main goal. It can be found in Figure 3 (a). Most models’ performance drops to 0 even under 20 iterations of the PGD attack within a maximum pixel perturbation of 5. All these observations show that the camera-based 3D object detection model is vulnerable to attacks that are even not that strong. Additionally, AutoAttack [1] is designed for image classification tasks, so, it’s significantly non-trivial to adapt it fully to 3D object detection tasks without large modification. We implement the AutoPGD attack, which is one of the components in AutoAttack to serve as a stronger attack method. We strictly follow the setting in the original AutoAttack paper [1] to add the following improvement:
>
> 1. Add momentum during the update, the momentum term is chosen to be 0.75, following [1]
> 2. Add a dynamic step size that adapts automatically with the optimization process
> 3. Add restart from the best points
> 4. Add exploration v.s. exploitation by using checkpoints $w_j$
>
> We fix $p_0=0$ and $p_1=0.22$ following the setting in [1], and the initial step size is $0.2\epsilon$, The results can be seen below. Besides, we also implement different kinds of attacks including FGSM and C&W attacks, the results can be seen in Answer 1 to Reviewer Vdtq.
>
> | Model   | NDS    | mAP    | mATE   | mASE   | mAOE   | mAVE   | mAAE   |
> | ------- | ------ | ------ | ------ | ------ | ------ | ------ | ------ |
> | DETR3D  | 0.1373 | 0.0522 | 0.9419 | 0.5150 | 1.0066 | 1.2340 | 0.4311 |
> | PETR    | 0.0880 | 0.0094 | 1.0032 | 0.6438 | 0.8749 | 1.2613 | 0.6482 |
> | FCOS3D  | 0.1562 | 0.0507 | 0.8930 | 0.4965 | 0.9639 | 1.0070 | 0.3380 |
> | PGD-Dev | 0.1700 | 0.0544 | 0.8494 | 0.5297 | 0.8157 | 1.2942 | 0.3775 |
> | BEVDet  | 0.0766 | 0.0203 | 1.0156 | 0.6403 | 1.0308 | 1.1250 | 0.6847 |
>
> We can see from the results that when applying AutoPGD, the 3D object detection models' performance is extremely bad even under only 10-step iteration, which further reveals the vulnerability of these models.
>
> ---
>
> **Reference**
>
> [1] [ICML'20] Reliable Evaluation of Adversarial Robustness with an Ensemble of Diverse Parameter-free Attacks

---

### Review · Reviewer_zozS · 2023-11-01

**Summary Of Contributions:**

The paper conducts a comprehensive evaluation of 3D object detection methods against adversarial attacks. The authors categorize the existing 3D object detection methods into a few classes and evaluate their robustness against pixel-level and patch-level adversarial attacks, respectively. The results points to four findings.

**Audience:**

Yes

**Broader Impact Concerns:**

Not applicable.

**Claims And Evidence:**

Yes

**Requested Changes:**

I am not familiar with the 3D object detection domain, so please correct me if I am wrong. The clean performance of the selected models seems relatively low. Most of the models have only 0.3-0.4 precision and detection scores. I am not sure if these are really state-of-the-art performances.  I would suggest the authors explicitly justify this or use models with better performance for evaluation. IMHO, it is more valuable to attack models with high performance rather than mediocre models.

The differences in attack performance are not that significant. To better support the arguments and findings of the paper, it would be more rigorous if the authors conduct the experiments multiple times, reporting the mean and stand devotion together with the paired t-test results.

**Strengths And Weaknesses:**

Strengths:
+ The paper clearly categorizes the adversarial attacks against 3D object detection.

+ The experiments are of a large scale and the corresponding findings are meaningful.

Weaknesses:

- The clean performance of the evaluated models is low.

- The differences in results are not that significant.

---

> ### Author Response · Authors · 2023-11-22
> **Re: Review of Paper1724 by Reviewer zozS**
>
> We are grateful for your thoughtful review and the opportunity to address the concerns raised regarding our paper.
>
> >  ***Question 1:** The clean performance of the evaluated models is low. not sure if these are really state-of-the-art performances. I would suggest the authors explicitly justify this or use models with better performance for evaluation.*
>
> **Answer 1**: In response to your query about the clean performance of our evaluated models, it's important to consider the specific challenges and benchmarks in the field of 3D object detection, particularly when using camera-based methods. The observed Mean Average Precision (mAP) metrics, ranging between 0.3 to 0.4, are indeed indicative of the current state-of-the-art for these types of models (open-sourced). This is especially true in the context of complex real-world datasets like nuScenes, where 3D detection faces additional hurdles like depth estimation and accurate 3D localization. We have selected models for our study that represent the forefront of current technological capabilities in this area. Furthermore, as shown in the table below, we list some of the most recent state-of-the-art models, along with their best open-source checkpoints. It’s important to note that BEVFormer and BEVDet, which are central to our study, have undergone further augmentation recently. Therefore, the best checkpoints currently are better than the checkpoints we used before. Nonetheless, their core architecture and pipeline remain consistent, which means our conclusion on these models can still potentially generalize to their newest versions.
>
> | Venue    | Model     | GitHub                                                 | Best Open-Source mAP | Best Open-Source NDS |
> | -------- | --------- | ------------------------------------------------------ | -------------------- | -------------------- |
> | arXiv'22 | Sparse4D  | [Link](https://github.com/linxuewu/Sparse4D)           | 0.439                | 0.538                |
> | ICCV'23  | HoP       | [Link](https://github.com/Sense-X/HoP)                 | 0.399                | 0.510                |
> | arXiv'22 | BEVDet    | [Link](https://github.com/HuangJunJie2017/BEVDet)      | 0.472                | 0.576                |
> | ECCV'22  | BEVFormer | [Link](https://github.com/fundamentalvision/BEVFormer) | 0.460                | 0.553                |
> | ICCV'23  | PETR      | [Link](https://github.com/megvii-research/PETR)        | 0.410                | 0.503                |
>
> >  ***Question 2**: The differences in attack performance are not that significant. To better support the arguments and findings of the paper, it would be more rigorous if the authors conduct the experiments multiple times, reporting the mean and stand devotion together with the paired t-test results.*
>
> **Answer 2**: In addressing your concern about the significance of differences in attack performance, we would like to re-emphasize that we undertook a comprehensive evaluation strategy. Our approach involved performing attacks under a variety of severities and averaging the results for a thorough analysis. Specifically for pixel perturbation attacks, we assessed model performance across a spectrum of iteration numbers: [1, 2, 3, 4, 5, 6, 7, 8, 9, 10, 20, 30, 40, 50], as shown in the Appendix. Consequently, the results for pixel perturbation attacks, as presented in Table 3 of our paper, represent an aggregate of 14 individual evaluations. This extensive testing methodology strengthens the rigor and validity of our findings.
>
> Furthermore, the primary results showcased in Table 2 are an aggregate average over all types of attacks and their respective severities. This encompasses a total of 31 different attack scenarios, calculated as the sum of 14 pixel-classification attacks, 9 pixel-localization attacks, 4 patch-classification attacks, and 4 patch-localization attacks. We believe this comprehensive evaluation not only enhances the robustness of our experiment results but also solidifies the reliability and validity of our conclusions.

---

### Review · Reviewer_Vdtq · 2023-11-09

**Summary Of Contributions:**

The work studies adversarial robustness of different camera-based 3D object detection approaches comprehensively. The 3D object detection approaches covered by the study include monocular approach and approaches based on bird's-eye-view (BEV) representations with or without depth estimation. The study considers pixel and patch based attack approaches using projected gradient descents (PGD) as well as black-box attack method. The work studies both clean performance, adversarial robustness, and the transferability of adversarial perturbations across models on the standard nuScenes dataset and draws critical and useful insights from the experiment results from the following perspectives: the use of BEV representations, explicit depth estimation, using temporal information, long-tail distribution of object classes, and model size.

**Audience:**

Yes

**Broader Impact Concerns:**

The work should not be exempt from having a Broader Impact Statement. Please see **Weaknesses** and **Requested Changes**.

**Claims And Evidence:**

Yes

**Requested Changes:**

**Requested Changes**

The work should include a Broader Impact Statement discussing the real-world threats of the studied adversarial settings, ethical concerns, and broader impacts of the study's results.


**Suggested Changes**

I would encourage the authors to include study results based on other standard adversarial attack methods like FGSM and Carlini & Wagner in their study.

**Strengths And Weaknesses:**

**Strength**
1. The problem studied by the work is interesting to the community of both 3D vision and adversarial machine learning and closely related to the latest research with safety-critical real-world applications like autonomous driving based on camera vision.
2. The camera-based 3D object detection approaches covered by the work are new and comprehensive.
3. The experiment study settings are comprehensive, considering different adversarial attack methods and different targets. The comprehensiveness of the experiment settings also make the conclusions of the work solid.
4. The study results are well-presented with clear descriptions of background knowledge and experiment settings, making the work very readable.

**Weakness**
1. The white-box adversarial attack approaches only include PGD. However, there are other standard adversarial attack approaches including fast gradient sign method [1] and Carlini & Wagner [2] with strengths and shortcomings different from PGD. It would make the study even more comprehensive and the conclusions more solid if the study can show how effective these methods are against the 3D object detection approaches.
2. Since study is closely related to safety-critical real-world problems, the work should include a dedicated part discussing the real-world threats of the studied adversarial settings, ethical concerns, and broader impacts of the study's results.


[1] Goodfellow, Ian J., Jonathon Shlens, and Christian Szegedy. "Explaining and harnessing adversarial examples." arXiv preprint arXiv:1412.6572 (2014).

[2] Carlini, Nicholas, and David Wagner. "Towards evaluating the robustness of neural networks." 2017 ieee symposium on security and privacy (sp). Ieee, 2017.

---

> ### Author Response · Authors · 2023-11-22
> **Re: Review of Paper1724 by Reviewer Vdtq**
>
> We extend our gratitude to Reviewer Vdtq for your comprehensive review and insightful feedback on our work. Your constructive criticism is highly valued and will undoubtedly aid in refining our research. In response to each of your observations, we have detailed the modifications and enhancements we plan to implement in our work.
>
> > ***Question 1**: The review notes the absence of other standard adversarial attack approaches, such as the Fast Gradient Sign Method (FGSM) and the C&W attack, which have distinct strengths and weaknesses compared to the Projected Gradient Descent (PGD).*
>
> **Answer 1**: We added the experiment results using FGSM and C&W Attack on these models: DETR3D, PETR, BEVDet, FCOS3D, and PGD-Det.
>
> - **Experiment Setting**
>
> In alignment with the adversarial setting outlined in our paper, we implemented FGSM attacks with an $\epsilon$ value of 5 under the $L_{\infty}$ norm. For the C&W attack, we calibrated the scale parameter $\lambda$ to a specific value of 10 to ensure a fair comparison. Both attacks were conducted using classification and localization as the objectives, consistent with the approach in our original manuscript. The detailed results of these additions are as follows:
>
> - FGSM Classification
>
> | Model       | NDS    | mAP    | mATE   | mASE   | mAOE   | mAVE   | mAAE   |
> | ----------- | ------ | ------ | ------ | ------ | ------ | ------ | ------ |
> | DETR3D      | 0.2259 | 0.1406 | 0.9066 | 0.4804 | 0.7946 | 0.9122 | 0.3498 |
> | PETR        | 0.1501 | 0.0710 | 0.9419 | 0.5616 | 0.9664 | 0.9048 | 0.4789 |
> | BEVDet | 0.1427 | 0.0603 | 1.0925 | 0.4855 | 1.1231 | 1.1421 | 0.3884 |
> | FCOS3D      | 0.1712 | 0.1104 | 0.9319 | 0.5034 | 1.0274 | 1.3144 | 0.4053 |
> | PGD-Dev     | 0.1848 | 0.0865 | 0.9440 | 0.5292 | 0.8091 | 0.9069 | 0.3949 |
>
> - FGSM Localization
>
> | Model       | NDS    | mAP    | mATE   | mASE   | mAOE   | mAVE   | mAAE   |
> | ----------- | ------ | ------ | ------ | ------ | ------ | ------ | ------ |
> | DETR3D      | 0.2441 | 0.2095 | 0.8805 | 0.4892 | 0.7957 | 1.1887 | 0.4413 |
> | PETR        | 0.1778 | 0.1273 | 0.9900 | 0.4739 | 1.0308 | 1.1836 | 0.3951 |
> | BEVDet | 0.1675 | 0.1249 | 0.9852 | 0.4984 | 1.0121 | 1.2842 | 0.4655 |
> | FCOS3D      | 0.1583 | 0.0942 | 0.9647 | 0.4982 | 0.9403 | 1.5566 | 0.4852 |
> | PGD-Dev     | 0.1606 | 0.0960 | 1.0325 | 0.5231 | 0.9541 | 1.0494 | 0.3963 |
>
> - C&W Classification
>
> | Model   | NDS    | mAP    | mATE   | mASE   | mAOE   | mAVE   | mAAE   |
> | ------- | ------ | ------ | ------ | ------ | ------ | ------ | ------ |
> | DETR3D  | 0.2765 | 0.2030 | 0.8686 | 0.4752 | 0.7202 | 0.8435 | 0.3419 |
> | PETR    | 0.1656 | 0.1167 | 0.9507 | 0.4833 | 1.0228 | 1.5613 | 0.4938 |
> | FCOS3D  | 0.1119 | 0.0730 | 1.0815 | 0.5873 | 1.0157 | 1.0151 | 0.6586 |
> | PGD-Dev | 0.1585 | 0.0924 | 1.0400 | 0.5003 | 0.8569 | 1.3132 | 0.5195 |
>
> - C&W Localization
>
> | Model   | NDS    | mAP    | mATE   | mASE   | mAOE   | mAVE   | mAAE   |
> | ------- | ------ | ------ | ------ | ------ | ------ | ------ | ------ |
> | DETR3D  | 0.2793 | 0.1461 | 0.8809 | 0.4635 | 0.7118 | 0.5647 | 0.3165 |
> | PETR    | 0.1681 | 0.0870 | 0.8943 | 0.4938 | 0.9151 | 1.3880 | 0.4513 |
> | FCOS3D  | 0.1217 | 0.0632 | 1.0194 | 0.5774 | 0.8748 | 1.4137 | 0.6470 |
> | PGD-Dev | 0.1485 | 0.0761 | 0.9929 | 0.5214 | 0.9067 | 1.3131 | 0.4745 |
>
> The new experiment results highly align with our conclusion. For instance, BEV-based detectors show better robustness under localization attacks. Additionally, it's interesting to see FGSM shows a strong attack performance given its simplicity.
>
> > ***Question 2**: Since the study is closely related to safety-critical real-world problems, the work should include a dedicated part discussing the real-world threats of the studied adversarial settings, ethical concerns, and broader impacts of the study's results.*
>
> **Answer 2**: In addressing the real-world threats of the studied adversarial settings, our research prioritizes identifying vulnerabilities in camera-based 3D detection models, focusing on digital adversarial attacks. We highlight the necessity of extending our findings to physical attack scenarios in future studies, given their potential to significantly impact autonomous driving systems. As for the ethical concerns, our work underscores the responsibility of developers to ensure the safety and reliability of these systems, particularly in the context of public spaces where they are more susceptible to adversarial manipulations. The broader impacts of our study emphasize the importance of incorporating comprehensive security measures in the development and deployment of such technologies, acknowledging the risk of societal harm, and advocating for a more secure and ethically responsible approach to technology implementation in safety-critical applications.

---

### Author Response · Authors · 2023-11-22
**General Response**

We sincerely thank our reviewers for devoting time to this review and offering valuable comments.

------

We are glad to see that the reviewers are acknowledging:

- "The experiments are large-scale and comprehensive" (Reviewer Vdtq, Reviewer zozS, Reviewer EhUz).
- "The camera-based 3D object detection approaches covered by the work are new and comprehensive" (Reviewer Vdtq).
- "The problem is interesting and closely related to the latest research with safety-critical real-world applications" (Reviewer Vdtq).
- "The problem is well-presented and clarified" (Reviewer Vdtq, Reviewer zozS, Reviewer EhUz)

------

We will polish the content, add more evidence and results, and clarify potential issues in the next revision. Specifically, we will include the following changes according to reviewers’ insightful comments:

- Description & Assessment
  - As suggested by Reviewer Vdtq, we added more discussion on the real-world threats of the studied adversarial settings, ethical concerns, and broader impacts of the study's results.
  - As suggested by Reviewer zozS, we added more justification on the model selection and the argument on the differences among different attacks.
- Experiments & Observation
  - As suggested by Reviewer Vdtq, we supplemented more results on attacks using FGSM and C&W attacks.
  - As suggested by Reviewer EhUz, we supplemented results using AutoAttack.

------

We will actively participate in the Author-Reviewer discussion session. Please don’t hesitate to let us know of any additional comments on the manuscript.

Last but not least, we thank the reviewers again for the time and effort devoted to this review.

---

### Decision · Action_Editor_tbHe · 2023-12-20

**Recommendation:** Accept as is

**Comment:**

The adversarial robustness of 3D point models is an interesting research problem studied by the authors. The paper's comprehensive and robust experimental study lends strong empirical support to its claims. Furthermore, the authors demonstrate a constructive response to critiques, effectively addressing concerns regarding the simplicity of attacks and the depth of analysis. All reviewers have recommended the acceptance of this paper.

**Audience:**

The findings of this paper would interest at least some individuals in TMLR's audience, given its interdisciplinary nature that intersects 3D vision and adversarial machine learning. The focus on the novel area of adversarial robustness in 3D point models presents fresh insights in a relatively new field, making it appealing to those following the latest developments in machine learning. Additionally, the paper's comprehensive and solid experimental work adds empirical value, despite some concerns regarding methodological originality.

**Claims And Evidence:**

This submission provides accurate and convincing evidence supporting its claims, particularly in its relevance to the 3D vision and adversarial machine learning communities, as well as its comprehensive and solid experimental study. However, concerns are raised regarding the originality of the methodology, as it primarily applies existing adversarial attack methods. But these were adequately addressed by the authors.